# Hand-Washing Video Dataset Annotated According to the World Health Organization's Hand-Washing Guidelines

Martins Lulla [1], Aleksejs Rutkovskis [1], Andreta Slavinska [1], Aija Vilde [1,2], Anastasija Gromova [1], Maksims Ivanovs [3], Ansis Skadins [3], Roberts Kadikis [3] and Atis Elsts [1,3,*]

[1] Medical Education Technology Centre, Riga Stradins University, Dzirciema iela 16, LV-1007 Riga, Latvia; martins.lulla@rsu.lv (M.L.); aleksejs.rutkovskis@rsu.lv (A.R.); andreta.slavinska@rsu.lv (A.S.); aija.vilde@rsu.lv (A.V.); anastasija.gromova@rsu.lv (A.G.)

[2] Department of Infectious Diseases and Hospital Epidemiology, Pauls Stradins Clinical University Hospital, Pilsonu Street 13, LV-1002 Riga, Latvia

[3] Institute of Electronics and Computer Science (EDI), Dzerbenes 14, LV-1006 Riga, Latvia; maksims.ivanovs@edi.lv (M.I.); ansis.skadins@edi.lv (A.S.); roberts.kadikis@edi.lv (R.K.)

* Correspondence: atis.elsts@edi.lv

**Abstract:** Washing hands is one of the most important ways to prevent infectious diseases, including COVID-19. The World Health Organization (WHO) has published hand-washing guidelines. This paper presents a large real-world dataset with videos recording medical staff washing their hands as part of their normal job duties in the Pauls Stradins Clinical University Hospital. There are 3185 hand-washing episodes in total, each of which is annotated by up to seven different persons. The annotations classify the washing movements according to the WHO guidelines by marking each frame in each video with a certain movement code. The intention of this "in-the-wild" dataset is two-fold: to serve as a basis for training machine-learning classifiers for automated hand-washing movement recognition and quality control, and to allow to investigation of the real-world quality of washing performed by working medical staff. We demonstrate how the data can be used to train a machine-learning classifier that achieves classification accuracy of 0.7511 on a test dataset.

**Dataset:** https://doi.org/10.5281/zenodo.4537209

**Dataset License:** CC-BY-SA

**Keywords:** hand-washing; hand movements; video dataset

## 1. Summary

In 2019 European Centre for Disease Prevention and Control (ECDC) and the World Health Organisation (WHO) declared curbing anti-microbial resistance as one of the global health priorities. The death toll from infections caused by multidrug-resistant bacteria reached 34,000 per year in Europe and 700,000 worldwide [1–3]. In 2020 the world faced an even greater global health crisis—COVID-19 pandemic caused by SARS-CoV-2. Along with other infection prevention and control (IPC) measures, hand hygiene has been a critical and low-cost safety measure required for preventing the spread and cross-transmission of both multidrug-resistant bacteria and SARS-CoV-2 [4]. It is crucial to perform hand hygiene correctly, following the WHO guidelines on the six key hand-washing moments and the appropriate amount of time [5]. However, the general public and even medical professionals worldwide often neglect these guidelines [6–9] despite perpetual educational campaigns [10]. The COVID-19 pandemic has accelerated this issue even further. To develop the institution and group-specific recommendations to improve compliance with the WHO guidelines, it is necessary to automate the quality control of hand hygiene.

To develop machine-learning classifiers for accurate hand-washing movement recognition, large-scale real-world datasets are necessary. Few examples of such datasets are openly available. The Kinetics Human Action Video Dataset [11] by Google contains 916 videos of washing hands. The Kaggle data science site provides a Hand Wash Dataset [12], a publicly available sample which has 292 videos labeled according to the WHO guidelines. The STAIR Actions dataset [13] consists of more than 100,000 videos, of which around 1000 are related to washing hands. However, the examples are limited in multiple ways: none of them has more than 1000 hand-washing videos, they do not focus on medical professionals, and only the Kaggle dataset provides labels according to the WHO guidelines.

To address these limitations, this paper presents a large-scale real-world dataset collected in summer 2020 in one of the largest hospitals in Latvia, the Pauls Stradins Clinical University Hospital.

## 2. Dataset Description

The dataset consists of video files along with their annotations in CSV and JSON formats. Table 1 presents an overview of the dataset.

**Table 1.** Dataset overview.

| Property | Value |
|---|---|
| Frame rate | 30 FPS |
| Resolution | 320 × 240 and 640 × 480 |
| Number of videos | 3185 |
| Number of annotations | 6690 |
| Total washing duration | 83,804 s |
| Movement 1–7 duration | 27,517 s |

### 2.1. Folder Structure

The files in the dataset are structured as follows:

```
DataSets
\- Dataset1
\- Videos
\- 2020-06-27_11-57-25_camera104.mp4
\- 2020-06-28_18-28-10_camera102.mp4
\- ...
\- Annotations
\- Annotator1
\- 2020-06-27_11-57-25_camera104.csv
\- 2020-06-27_11-57-25_camera104.json
\- ...
\- Annotator2
\- 2020-06-27_11-57-25_camera104.csv
\- 2020-06-27_11-57-25_camera104.json
\- ...
\- Dataset2
\- Videos
\- ...
\- Annotations
\- ...
\- Dataset3
\- Videos
\- ...
\- Annotations
\- ...
```

```
...
summary.csv
statistics.csv
```

Each video file has annotations from one or more annotators. For convenience, two annotation formats are included, although most of the information in the CSV and JSON files is overlapping. A video file `name`.mp4 has annotations in both `name`.csv and `name`.json. Additionally, several files describing aggregate information are present. The file `summary.csv` contains a summary of the dataset, and the file `statistics.csv` contains the main metrics for each hand-washing episode in the dataset.

### 2.2. Annotations

Each frame in each video is annotated with the following information: (1) whether hand-washing was visible in the frame, and (2) which of the WHO movements, if any, did the hand-washing corresponded to.

Each CSV file contains three columns. The first column is `frame_time`—the time in seconds of the frame in the video—and the second and third columns, called `is_washing` and `movement_code`, contain the movement annotations. Each JSON file contains a dictionary with several keys. The data under the "`labels`" key contains movement annotations for each frame. The other keys contain supplementary information about the quality of the hand-washing performed in the video. They are: "`is_ring_present`" "`is_armband_present`" "`is_long_nails_present`", and contain information about whether the person washing hands has a ring, an armband or watch, and long (artificial) nails.

The movement codes in the annotations correspond to specific hand-washing movements as defined by the WHO guidelines [5], described in Table 2.

**Table 2.** Movement codes.

| Code | Movement |
|:---:|---|
| 1 | Hand-washing movement—Palm to palm |
| 2 | Hand-washing movement—Palm over dorsum, fingers interlaced |
| 3 | Hand-washing movement—Palm to palm, fingers interlaced |
| 4 | Hand-washing movement—Backs of fingers to opposing palm, fingers interl. |
| 5 | Hand-washing movement—Rotational rubbing of the thumb |
| 6 | Hand-washing movement—Fingertips to palm |
| 7 | Turning off the faucet with a paper towel |
| 0 | Other hand-washing movement |

According to our annotation guidelines presented for the annotators:

- Code from 1 to 6 is used to denote a correctly performed hand-washing movement that corresponds to one of the WHO movements.
- Code 0 is used to denote both WHO washing movements that are not performed correctly, and washing movement that have not been defined by the WHO.
- Code 7 is used to denote the process of correctly terminating the hand-washing episode. Specifically, in order to use the code 7 in the video, we require that the person who washes hands takes a paper towel, dries their hands with it, and then closes the faucet with the towel.
- Frames that do not have hand-washing depicted are labeled as such (`is_washing` set to zero). The movement code should be ignored for such frames.

### 2.3. Quality Issues

To increase the reliability of the annotations, the majority of files in the dataset are labeled by more than one annotator (Figure 1). Overall, there is a reasonably good match between the annotators. For example, frames that are annotated by two annotators have 91.23% agreement on the `is_washing` field. Those of the frames that have both annotators

setting `is_washing` to one, further have 90.06% agreement in the movement code. We have identified some reasons for a mismatch:

- Short-term disagreement between the labels typically exists at the points when the washing movement is changed.
- Movements 1 and 3 look quite similar and can be hard to distinguish when filmed at an angle.
- The interpretation of what constitutes movement 7 has been different between the different reviewers.
- Some of the videos have low light levels.

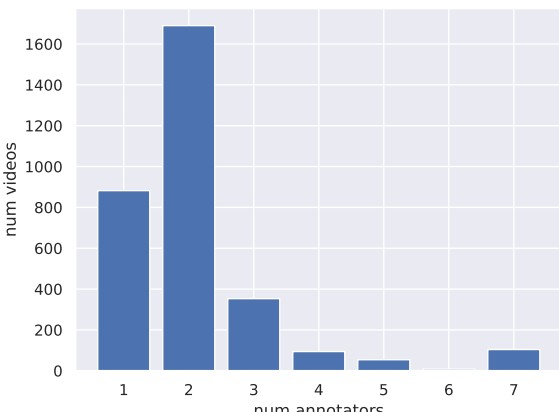

**Figure 1.** Number of annotators per video.

### 3. Methods

The videos were recorded on either AirLive IP cameras or Axis IP cameras. They were saved by Raspberry Pi 4 devices, which had the cameras attached over Ethernet cables. The cameras were deployed in nine different locations simultaneously, with one location corresponding to one sink. In total there were 12 cameras, and some of the Raspberry Pi devices had more than one camera attached. This enabled us to record hand-washing in a single sink simultaneously from different angles.

The locations where the cameras were deployed included a hospital neurology unit, surgery unit, an intensive care unit, and other hospital units in the Pauls Stradins Clinical University Hospital. In the dataset, the locations are anonymized. In case location information is required for your research, contact us for the list of locations and their correspondence to the directories in the dataset.

The cameras recorded all continuous movements within their field of view. To filter out short-term movements (e.g., a person passing by), recording was only started in the case when motion was detected for 3 s continuously. As a result, the videos in the dataset may have up to first 3 s of each hand-washing episode missing. Videos shorter than 20 s were not saved by the recording system to minimize the number of false-positive motion detections.

The recorded data was manually collected to a central server at the Riga Stradins University by bringing in the SD cards from the Pi's and uploading their data. Subsequently, annotators were given access to the video files on the server, and asked to label the files using a Python OpenCV application that we had developed for the task. The annotators pre-filtered the files to remove videos that did not include an actual hand-washing episode. Each annotator did this independently, based on our guidelines. As a result, some files have been filtered out by one annotator, but not by other ones. In the final dataset, there are instances when an annotation for a video is present in e.g., `Annotator1`, but not in `Annotator2`. The annotators are anonymized in the final dataset, and the folder `Annotator1` in one part of the dataset is not necessarily annotated by the same person as the folder `Annotator1` in a different part of the dataset.

## 4. Application Example

The intention of this dataset is two-fold: first, to serve as a basis for training machine-learning classifiers for automated hand-washing movement recognition and quality control; second, to allow investigation of the real-world quality of washing performed by working medical staff.

To demonstrate the first application, we trained on video data and annotations MobileNetV2 [14], a neural network classifier available in in Keras [15], a high-level deep learning API for the deep learning tensor library TensorFlow [16]. We aimed to recognize the Movements 1 to 6 as defined by the WHO, and to distinguish them from the movement 0. As to the movement 7, while it is important for clinical outcomes, is not relevant to our machine-learning goals here, so was treated as the movement 0 for the classification purposes.

We started by partitioning the dataset in two portions, *test* and *train and validation* (10% and 90%, respectively) to ensure that frames from test videos were kept separately from the frames from the training and validation datasets. We subsequently extracted frames from the videos and saved them as JPEG files. As the class 0 is over-represented in the data, we only saved 20% of the JPEG files corresponding to this class. Further data processing consisted of resizing the images files to $224 \times 224$ pixels, which is the standard input size for MobileNetV2 implementation in Keras, and applying random flipping and rotations by 20 degrees to them to augment the data.

To obtain the first results quickly, we used MobileNetV2 model pre-trained on the Imagenet dataset [17] and trained the model for 10 epochs using Adam optimizer with learning rate of 0.008 and categorical loss function (Figure 2); no more refined hyperparameter tuning was done. As a result, we achieved classification accuracy of 0.7511 on the test dataset.

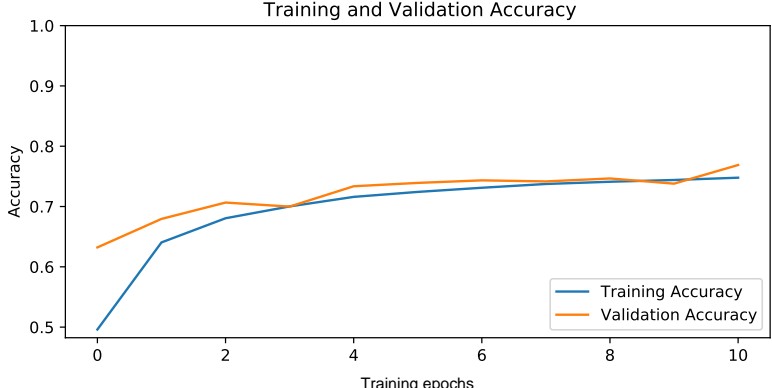

**Figure 2.** Training and validation accuracy of MobileNetV2 model.

Python scripts for preprocessing the dataset and training the MobileNetV2 model are available at https://github.com/edi-riga/handwash, accessed on 22 March 2021. More extensive evaluation of machine-learning classifiers evaluated on the data is available in [18].

**Author Contributions:** Conceptualization, A.E., R.K., A.S. (Andreta Slavinska), M.L., A.R.; methodology, A.E., M.I., R.K., A.S. (Andreta Slavinska), M.L., A.R., A.V., A.G.; software, M.I., A.S. (Ansis Skadins), A.E.; validation, M.I., A.S. (Ansis Skadins), A.E.; data curation, M.L., A.E., A.R.; writing—original draft preparation, M.L., A.R., A.E., M.I.; visualization, A.E., M.I.; supervision, A.E., R.K.; project administration, A.S. (Andreta Slavinska); funding acquisition, A.S. (Andreta Slavinska), A.E. All authors have read and agreed to the published version of the manuscript.

**Funding:** This research was funded by the Ministry of Education and Science, Republic of Latvia, project "Integration of reliable technologies for protection against COVID-19 in healthcare and high-risk areas", project No. VPP-COVID-2020/1-0004.

**Institutional Review Board Statement:** The study was conducted according to the guidelines of the Declaration of Helsinki, and approved by the Ethics Committee of Riga Stradins University (protocol code Nr. 6-1/08/10, date of approval 23 July 2021).

**Informed Consent Statement:** Informed consent was obtained from all subjects involved in the study.

**Data Availability Statement:** Data is available at https://doi.org/10.5281/zenodo.4537209, accessed on 22 March 2021.

**Acknowledgments:** We thank the Pauls Stradins Hospital for allowing us to collect data in their premises. We also extensively thank all who participated in the labor-intensive task of annotating the data.

**Conflicts of Interest:** The authors declare no conflict of interest.

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
