# Peer review of "Hand-Washing Video Dataset Annotated According to the World Health Organization’s Hand-Washing Guidelines"

_data, 2021_

Round 1

Reviewer 1 Report

The manuscript entitled “Hand Washing Video Dataset Annotated According to the World Health Organization’s Handwashing Guidelines” presents an issue associated with the COVID-19 pandemic.

Abstract:

  • Abstract must be expanded. More information associated with the results should be presented.
  •  

Introduction section:

  • The introduction section must be expanded. In this section Authors should present the information associated with COVID-19 pandemic
  • The introduction section must be expanded. In this section Authors should present the information associated with COVID-19 pandemic protection behaviors (Hand Washing). This section should be briefly presented – what do we know and what is the background for this study. Some detailed information about other studies are necessary. Authors should present their Introduction based on accurate literature – if they conducted their study for the COVID-19 epidemic, they should find adequate references associated with personal Protective Behaviors, as there are a lot of such studies published – e.g. for the issue of hand hygiene: https://www.ncbi.nlm.nih.gov/pmc/articles/PMC7267118/; https://www.ncbi.nlm.nih.gov/pmc/articles/PMC7459707/;  https://www.ncbi.nlm.nih.gov/pmc/articles/PMC7195203/. The good background should present the history of problem, the current knowledge and scientific "gap", and then authors should present how their study could fill this gap to justify the study.
  • The justification of the study should be presented.

Material and methods section:

  • Line 107 – “nine different locations simultaneously” – please specify the criteria for choosing this locations
  • Lines 120-121 – “Videos shorter than 20 seconds were not saved by the recording system in order to minimize the number of false-positive motion detections.” Some hand washing procedure may take less than 20 seconds – so this procedure could causes some bias.
  • The obtained results should be discussed, especially in the context of “class 0”. Any possible solution of this problem should be presented. Moreover, the possible practical application of this solution should be presented.

Author Response

See the attached document for our response.

Reviewer 2 Report

The authores put a lot of effort into the study. Originality is given and the present manuscript and dataset deal with an important topic. This project may help to reinforce the issue of correct handwashing.

Author Response

We thank for the comments.

Reviewer 3 Report

The paper presents a good source of hand wash dataset

Author Response

We thank for the comments.

Round 2

Reviewer 1 Report

I have no further comments